# Composite Polymer Electrolytes Based on (PEO)_4_CF_3_COOLi and Multi-Walled Carbon Nanotube (MWCNT)

**DOI:** 10.3390/polym15010049

**Published:** 2022-12-23

**Authors:** Miguel I. Delgado-Rosero, Nori M. Jurado-Meneses, Ramiro Uribe-Kaffure

**Affiliations:** Physics Department, University of Tolima, Ciudad Universitaria, Altos de Santa Helena, Ibague 730006, Colombia

**Keywords:** polymeric electrolytes, MWCNTs, DSC, FTIR, impedance spectroscopy

## Abstract

The addition of nanoparticles as fillers has a significant influence in modifying the dynamic conditions and avoiding crystallization in polymer composites. In this work, (PEO)_4_CF_3_COOLi electrolyte and multi-walled carbon nanotubes (MWCNTs) were used to prepare composites by a solution method. The formation of the new composite was evidenced by the experimental results obtained from DSC analysis and infrared spectroscopy (FTIR). The impedance spectroscopy analysis shows a notable decrease in the resistance, which is attributed to an interaction between the oxygen of the polymer and the Li^+^ cations of the salt, and the interactions between the electrolyte and the MWNTs. Values of dc conductivity of 8.42 × 10^−4^ S cm^−1^ at room temperature are obtained at a concentration of 2.0 wt.% MWCNT in the whole electrolyte. The results indicate that membranes can be used in technological devices such as batteries and gas or moisture sensors.

## 1. Introduction

The steady increase in demand for better and safer batteries for use in electric vehicles, mobile devices, and energy storage has driven research towards the development of new solid polymer electrolytes (SPEs). Among the main properties required for applications of SPEs in devices for technological use are high ionic conductivity, wide thermal and electrochemical window, and stability in different environmental conditions of temperature and humidity [1,2]. However, the relatively lower ionic conductivity of conventional liquid electrolytes is the main drawback of SPEs to achieve adequate practical utility.

Among the great variety of SPEs, composites formed by polymers, salts, and nanoparticles have received considerable attention in recent years due to their considerably improved electrical, thermal, mechanical, and optical properties concerning their precursors [3,4]. This improvement in the properties of SPEs is obtained thanks to the combination of phases and due to an appropriate interaction between the components. High conductivities in these systems are achieved due to high amorphous phase fractions or by combinations of liquid and solid phases, in which the nanoparticles used as fillers have a high influence on modifying the dynamic conditions and avoiding crystallization.

In particular, carbon nanotubes (CNTs) like single-walled carbon nanotubes (SWCNTs) or multi-walled carbon nanotubes (MWCNTs), which have good mechanical strength, low density, high electrical conductivity, and high thermal stability, can be used to modify the electric and mechanics properties of polymeric matrices [5,6,7,8,9].

Although some combinations of polymers with CNTs are used in systems that involve electronic conductivity [5,7,10], many applications in batteries and sensors require ionic conduction. In fact, it was shown that when CNT are added to solid electrolytes they can improve the ionic conductivity of the system [11,12]. For instance, some SPEs with lithium ions have shown an increase in ionic conductivity with the incorporation of CNTs, due to intercalation, adsorption, and diffusion of lithium ions into the CNT structure. A high affinity between the electronic cloud of CNTs and Li^+^ ions increases the fraction of free ions and provides low lattice energy pathways for ionic migration [13]. However, there is a possibility that using (CNTs) as fillers in polymers may cause a short circuit between the electrodes of electrochemical devices.

Many different methodologies were used to deposit CNTs and other conductive fillers in polymeric matrices; however, the dispersion of these fillers was a difficult task. For instance, Hammad R. Khaliden and colleagues reported a simple spray dispersion process of CNTs between two polymer layers, reducing the electrical resistance from 12 MΩ to 4 kΩ [10,14,15]. In this work, polymer composites were synthesized, by solution, from the (PEO)_4_CF_3_COOLi electrolyte added with different amounts of MWCNT, in order to improve ionic conductivity in the electrolyte. The electrical, thermal, and structural properties were studied by impedance spectroscopy, differential scanning calorimetry and infrared spectroscopy. The results indicate that membranes can be used in technological devices such as batteries and gas or moisture sensors.

## 2. Materials and Methods

### 2.1. Materials

The following materials were used for sample preparation: polyethylene oxide (PEO), multiwalled carbon nanotube (MWCNT), used as filler, and lithium trifluoroacetate (CF_3_COOLi). All materials were purchased from Aldrich. Basic properties of raw materials are consigned in Table 1, Table 2 and Table 3. Acetonitrile was used as common solvent.

### 2.2. Sample Preparation

The membranes were prepared by solution. Initially, PEO polymer and CF_3_COOLi salt were held for 24 h in a desiccator with silica gel to remove surface moisture. Subsequently, these compounds were dissolved separately in acetonitrile with magnetic stirring for 4 h at room temperature. Then, the two solutions were mixed in a 4:1 ratio (EO:Li) and stirred for 14 h. When the mixture acquired viscous liquid properties, amounts of MWCNT were added at concentrations of 0.0%, 0.3%, 0.6%, 2.0%, 4.0%, 8.0%, and 12.0%, relative to the total mass of the PEO + salt electrolyte. Then, the whole mixture was sonicated for 30 min for homogeneous dispersion of MWCNT. Finally, the obtained viscous liquid was poured in a glass petric dish and left in a low humidity environment (<20% RH) to evaporate the solvent. After approximately 24 h, mechanically stable membranes were obtained. We have standardized the process above for the synthesis of this type of electrolyte [16].

### 2.3. Characterization Techniques

The obtained membranes were analyzed by impedance spectroscopy measurements with a HIOKI 3532-50 LCR analyzer, with Pt/membrane/Pt electrode configuration, for a frequency range from 50 Hz to 5 MHz (which is the working range of the device) and a temperature range from 263 K to 338 K (below the melting point of crystalline PEO).

Thermal analyses were performed by differential scanning calorimetry measurements, with a Setaram DSC evo 131 calorimeter, in a temperature range from 176 K to 430 K, at 10 K/min (heating rate used in many studies of polymeric systems [17,18]); above this temperature, no further transitions are observed. Aluminum sample holders of 30 μL and high-purity nitrogen purge gas were used.

Fourier transform infrared (FTIR) measurements were obtained with a Jasco FT/IR 6220 Fourier Transform Infrared spectrometer in the range of 4000 cm^−1^ to 400 cm^−1^ with a maximum resolution of 0.25 cm^−1^.

## 3. Results and Discussion

Figure 1 shows DSC thermograms between 176 and 238 K for the polymeric solid electrolyte (PEO)_4_CF_3_COOLi and different combinations of this electrolyte with MWCNT in weight percent, (PEO)_4_CF_3_COOLi + *x*% MWCNT (*x* = 0.0%, 0.3%, 0.6%, 2.0%, 4.0%, 8.0%, and 12.0%). The step observed in each case is characteristic of the glass transition temperature (T_g_) in these systems [19,20]. It is observed that for concentrations up to 2.0% MWCNT, the step assigned to the T_g_ shifts towards lower temperatures. This shift is related to an increase in the amorphous phase fraction of the new composite. However, for concentrations above 2.0% MWCNT, the T_g_ shifts towards higher temperatures. Perhaps the increase of MWNCT above 2% might generate more nucleation sites that improve crystallinity [21]. The T_g_ values obtained from the DSC thermograms are reported in Table 4.

DSC thermograms between 302 and 430 K are shown in Figure 2. The thermogram of a pure PEO membrane shows an endothermic anomaly around 345 K associated with the melting of the crystalline phase of the polymer [22,23]. If the polymer is combined with CF_3_COOLi, a new endothermic anomaly is observed around 402 K; this anomaly is associated with melting of the complex formed between the polymer and the salt [24,25]. When the (PEO)_4_CF_3_COOLi electrolyte is added with different concentrations of MWCNT, the endothermic anomalies corresponding to the melting of the crystalline phase of PEO and the melting of the complex change in both temperature and enthalpy. This would indicate changes in the crystalline phase fraction of both PEO and the complex formed between PEO and CF_3_COOLi.

As seen in Table 5, when the MWCNT concentration is increased for values up to x = 2.0 wt.%, a decrease in both T_m_ and ∆H values is observed. For higher concentrations, x > 2.0 wt.%, an increase in T_m_ values (corresponding to PEO melting, 1st anomaly in Table 5) is observed. The latter would indicate the formation of MWCNT clusters and polymer chains in the composite and, therefore, an increase in the crystalline phase.

Figure 3 contains the FTIR spectra for PEO, CF_3_COOLi, and the different MWCNT concentrations in the solid electrolyte (PEO)_4_CF_3_COOLi, in the range between 670 and 1310 cm^−1^. For CF_3_COOLi, the energy band observed at 727 cm^−1^ (a in the Figure 3) is assigned to the CF_3_ symmetric bending mode of CF_3_COOLi [3,26,27]; this band shifts slightly towards a lower wavenumber (722 cm^−1^) when CF_3_COOLi is combined with PEO and with the different MWCNT contractions. The peak observed at 797 cm^−1^ (b in the Figure 3) is assigned to the COO scissor vibrational mode [26,28].

In the PEO spectrum, the peak observed at 857 cm^−1^ (c in the Figure 3) corresponds to CO stretching vibrations [11]; these peaks shift slightly to the left in the (PEO)_4_CF_3_COOLi electrolyte and the different combinations of the electrolyte with MWCNT. The peak observed at 960 cm^−1^ (d in the Figure 3) is assigned to the asymmetric stretching of CH_2_ [11] and remains unchanged in all cases. On the other hand, the peak observed at 1076 cm^−1^ (e in the Figure 3) is assigned to the asymmetric stretching of C-O-C [11]. However, when PEO is combined with CF_3_COOLi and with different MWCNT concentrations, this latter peak shifts to the left, indicating the interaction of Li ions with oxygen atoms of the polymer to form the (PEO)_4_CF_3_COOLi complex. In turn, the peak corresponding to the CH_2_ symmetric stretching vibrational mode is observed at 1242 cm^−1^ (h in the Figure 3) [11]; these peak shifts slightly to the left (1240 cm^−1^) in all systems.

Similarly, the peak observed at 1295 cm^−1^ (i in the Figure 3) in the PEO spectrum is assigned to asymmetric CH_2_ torsion [29]. This peak is found shifted to the left in the electrolyte and the different combinations with MWCNT.

Figure 4 contains the FTIR spectra for PEO, CF_3_COOLi, and the different MWCNT concentrations in the solid electrolyte (PEO)_4_CF_3_COOLi, in the range between 1312 and 3000 cm^−1^. The peak observed at 1358 cm^−1^ (j in the Figure 4) is assigned to bending of CH_2_ [29], while the peak observed at 1470 cm^−1^ (k in the Figure 4) is assigned to bending vibration of CH [29]; the latter remains unchanged in the electrolyte and in the combinations of the electrolyte with MWCNT.

On the other hand, relative to the CF_3_COOLi spectrum in Figure 3, the broad peaks observed at 1146 cm^−1^ and 1202 cm^−1^ are assigned to the asymmetric and symmetric stretching vibrational modes, respectively (f and g in Figure 3). In turn, the peak observed at 1673 cm^−1^ (l in the Figure 4) [17], assigned to the asymmetric stretching of COO, shifts slightly to the right (1686 cm^−1^) in the electrolyte and the combinations of this electrolyte with MWCNTs, corroborating the complex formation between PEO and CF_3_COOLi.

Figure 5 shows a zoom of the spectrum between 1550 and 1800 cm^−1^, where new shoulders are observed at 1655 cm^−1^ and 1702 cm^−1^ (m and n in the Figure 5); these are assigned to free ions responsible for the conductivity and ionic aggregates respectively [17,30].

Figure 6 shows the Nyquist plots at room temperature (298 K), obtained by impedance spectroscopy for PEO and (PEO)_4_CF_3_COOLi + *x*% MWCNT systems. In the diagrams of PEO and (PEO)_4_CF_3_COOLi, a straight line can be observed on the right side of the impedance spectrum (low frequencies) followed by a semicircle on the left side of the spectrum (high frequencies). The resistance (R) in these polymeric systems can be obtained by projecting the point where the line meets the semicircle to the Z’ axis. The resistance values show a considerable decrease when PEO and CF_3_COOLi are combined, which confirms the formation of the polymer electrolyte (PEO)_4_CF_3_COOLi and supports the DSC and FTIR results. In the Nyquist plots corresponding to the combinations of the (PEO)_4_CF_3_COOLi electrolyte with different MWCNT concentrations, the characteristic line of the double-layer capacitance in blocking electrodes is observed, but only in the beginning of the semicircle. In these cases, the resistance is even lower due to the interactions between the electrolyte and the MWNTs.

The values of the real conductivity as a function of frequency were obtained using the relation
(1)$$\sigma \prime \left(\omega \right)=\frac{l}{A}\frac{Z\prime }{Z^{\prime 2}+Z^{\prime \prime 2}},$$
where *l* is the membrane thickness, *A* is the electrode area, *Z′* is the real impedance, and *Z″* is the imaginary impedance.

In Figure 7, which shows diagrams of real conductivity as a function of frequency (Bode diagrams) for different temperatures, the real conductivity is higher at higher temperatures. This behavior, observed in polymeric solid electrolytes or polymer composites, is due to the agitation of the polymer chains and indicates that ionic conduction in these systems is a thermally activated process. Two regions can be differentiated in these diagrams. One low-frequency region in which an increase of the real conductivity is evidenced. This is due to the polarization of the electrodes by the double-layer capacitance at the interface of the blocking electrodes and the electrolyte. In a second region, where the real conductivity remains constant as the frequency increases (plateau), behavior is associated with the movement of long-range ions (hopping process) [31]. The conductivity value in the latter region is assumed to be the value of the dc membrane conductivity.

Figure 8 shows plots of the dc conductivity as a function of the inverse of temperature for different MWCNT concentrations in the (PEO)_4_CF_3_COOLi electrolyte. For all systems, it is observed that the dc conductivity increases with increasing temperature following the Vogel Tamman Fulcher (VTF) equation [18,32]:(2)$$\sigma_{dc}=\sigma_{0}\exp\left(\frac{E_{A}}{k_{B}\left(T-T_{0}\right)}\right),$$
where *σ_dc_* is the conductivity obtained in the plateau, *σ*_0_ is the pre-exponential factor, *E_A_* is a parameter known as pseudo-activation energy, *k_B_* is the Boltzman constant, *T* is the absolute temperature, and *T*_0_ stands for the temperature (below the glass transition temperature) at which the free volume, responsible for ionic conduction, disappears, and the ionic conductivity cancels out. Table 6 shows the parameters for the fitting to Equation (2).

## 4. Conclusions

We synthesized new composites by the addition of MWCNT into the polymeric solid electrolyte (PEO)_4_CF_3_COOLi. The formation of the composite was evidenced by the experimental results. Analysis of the results reveals a direct correlation between the increase in ionic conductivity, formation of new bonds, and changes in the percentage of the amorphous phase.

The higher value of ionic conductivity (8.42 × 10^−4^ S cm^−1^) was reached in the (PEO)_4_CF_3_COOLi + 2.0% MWCNT composite. For MWCNT concentrations above 2%, nucleation sites might generate increases in the crystallinity and low ionic conductivity values.

The ionic conductivity reached a value that is an order of magnitude higher than the conductivity of the (PEO)_4_CF_3_COOLi electrolyte. This conductivity value suggests that this system can be used as an electrolyte in electrochemical devices.

## Figures and Tables

**Figure 1 polymers-15-00049-f001:**
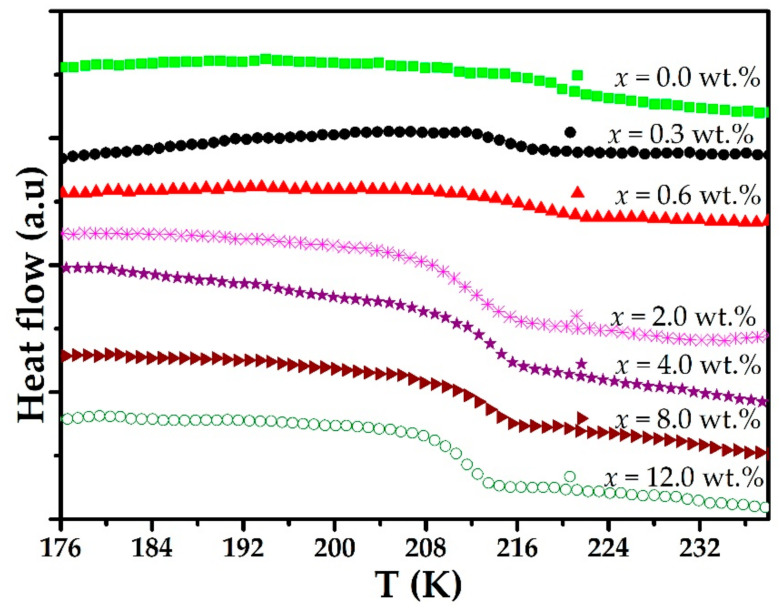
DSC thermograms for (PEO)4CF3COOLi + *x*% MWCNT composite (*x* = 0.0%, 0.3%, 0.6%, 2.0%, 4.0%, 8.0%, and 12.0%).

**Figure 2 polymers-15-00049-f002:**
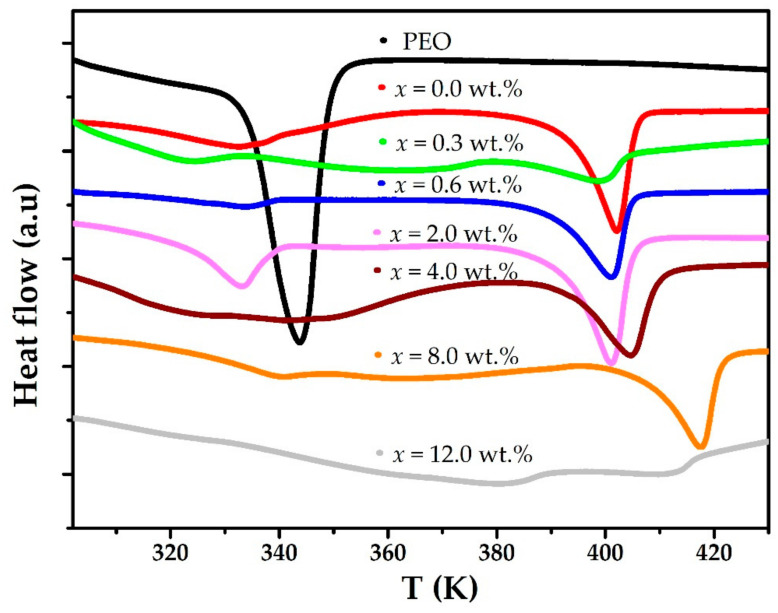
DSC thermograms between 302 and 430 K for PEO, the (PEO)_4_CF_3_COOLi electrolyte, and different combinations of (PEO)_4_CF_3_COOLi with MWCNT.

**Figure 3 polymers-15-00049-f003:**
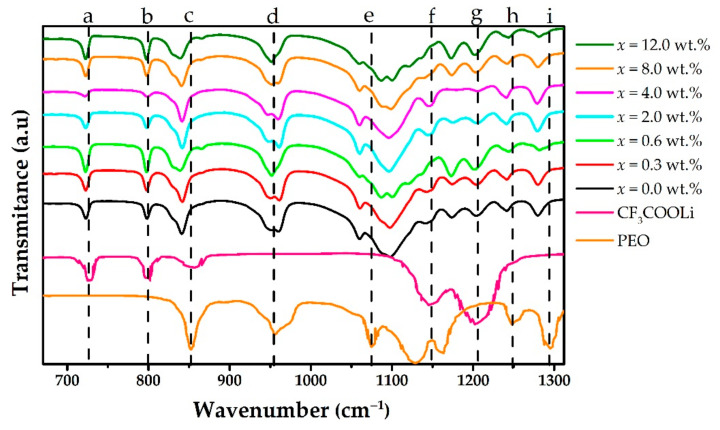
FTIR spectra for PEO, CF_3_COOLi, and different concentrations of MWCNT in the (PEO)_4_CF_3_COOLi electrolyte, for wavenumber ranging from 670 to 1312 cm^−1^. The dotted vertical lines stand for some wavenumber: a = 727 cm^−1^, b = 797 cm^−1^; c = 857 cm^−1^; d = 960 cm^−1^; e = 1076 cm^−1^; f = 1146 cm^−1^; g = 1202 cm^−1^; h = 1242 cm^−1^; i = 1295 cm^−1^.

**Figure 4 polymers-15-00049-f004:**
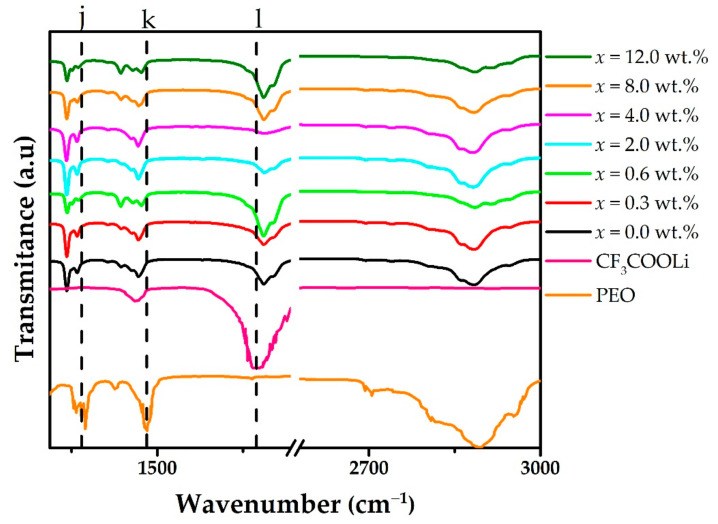
FTIR spectra for PEO, CF_3_COOLi, and different concentrations of MWCNT in the (PEO)_4_CF_3_COOLi electrolyte, for wavenumber ranging from 1312 to 3000 cm^−1^. The dotted vertical lines stand for some wavenumber: j = 1358 cm^−1^, k = 1470 cm^−1^; l = 1673 cm^−1^.

**Figure 5 polymers-15-00049-f005:**
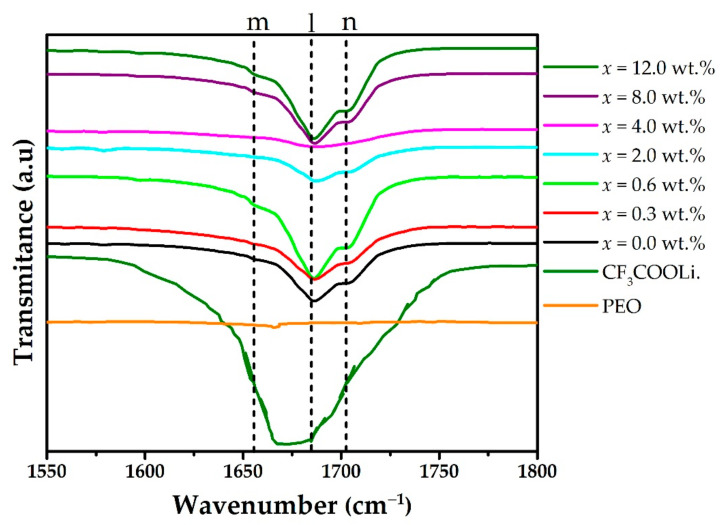
FTIR spectra for PEO, CF_3_COOLi, and different concentrations of MWCNT in the (PEO)_4_CF_3_COOLi electrolyte, for wavenumber ranging from 1550 to 1800 cm^−1^. The dotted vertical lines stand for some wavenumber: m = 1655 cm^−1^, l = 1673 cm^−1^; n = 1702 cm^−1^.

**Figure 6 polymers-15-00049-f006:**
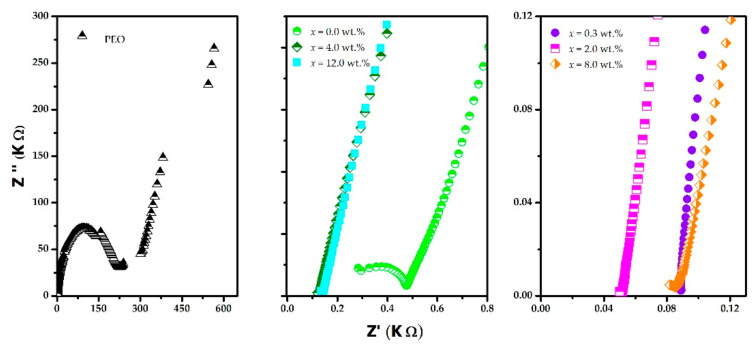
Nyquist diagrams for PEO and for (PEO)_4_CF_3_COOLi electrolyte with different combinations of MWCNT.

**Figure 7 polymers-15-00049-f007:**
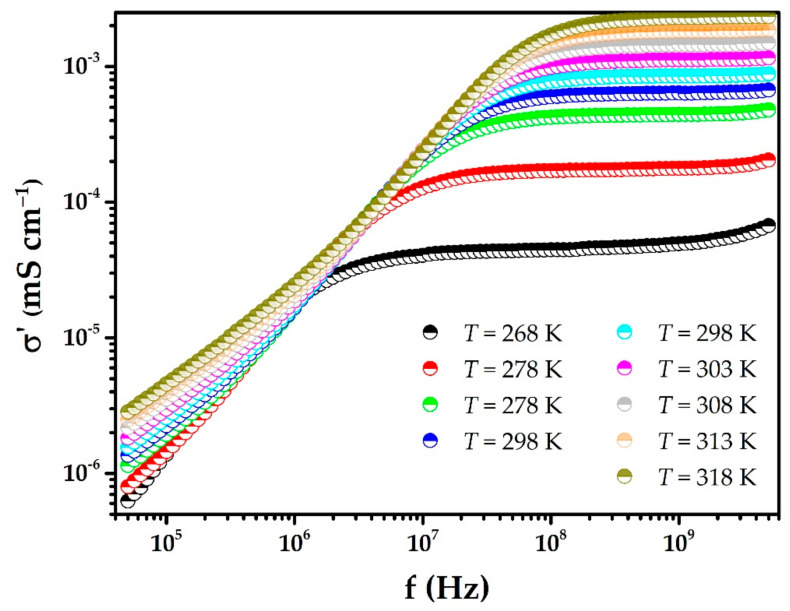
Real conductivity as a function of frequency for a (PEO)_4_CF_3_COOLi + 2.0% MWCNT membrane at different temperatures.

**Figure 8 polymers-15-00049-f008:**
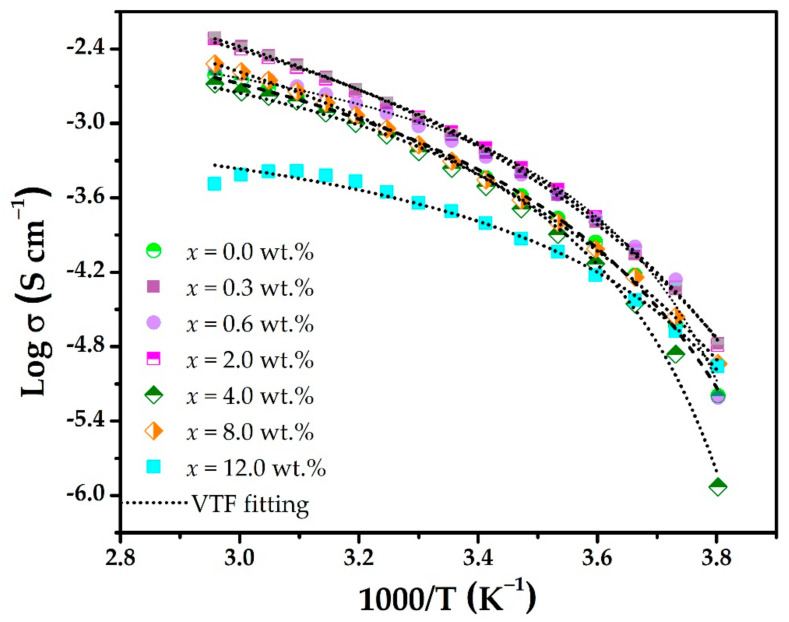
Logarithm of the conductivity as a function of the inverse of temperature for (PEO)_4_CF_3_COOLi + *x*% MWCNT composites.

**Table 1 polymers-15-00049-t001:** Basic properties of PEO.

Property	Value
Molecular weight average	5,000,000
Melting point	65 °C
Glass transition	−65 °C
Density	1.21 g/cm^3^

**Table 2 polymers-15-00049-t002:** Basic properties of CF_3_COOLi.

Property	Value
Molecular weight	119.96 g/mol
Purity	95%
Melting point	250 °C
Density	1.743 g/cm ^ 3 ^

**Table 3 polymers-15-00049-t003:** Basic properties of MWCNT.

Property	Value
Density	2.1 g/cm^3^
Outer diameter	7–15 nm
Length	0.5–10 μm
Melting point	3652–3697 °C

**Table 4 polymers-15-00049-t004:** T_g_ values obtained at the midpoint of the DSC thermogram step for the (PEO)_4_CF_3_COOLi electrolyte and for the combinations with MWCNT.

Concentration (wt.%)	Glass Transition Temperature T_g_ (K)
0.0	219.5
0.3	215.2
0.6	214.4
2.0	210.7
4.0	212.7
8.0	213.5
12.0	211.3

**Table 5 polymers-15-00049-t005:** Enthalpies (∆H) and melting temperatures (T_m_) for the first and second endothermic anomalies in DSC heating sweeps.

x% MWCNT	1st Anomaly		2nd Anomaly	
	∆H (J/g)	T_m_ (K)	∆H (J/g)	T_m_ (K)
PEO	134.9	344.0	-	-
0.0	44.8	336.2	84.5	402.1
0.3	25.9	334.0	16.8	399.0
0.6	20.6	333.6	67.2	401.1
2.0	18.2	332.9	64.7	401.1
4.0	92.6	341.1	49.4	404.7
8.0	4.8	350.2	48.0	403.4
12.0	10.4	367.2	4.6	397.6

**Table 6 polymers-15-00049-t006:** Values for fitting of parameters in Equation (2).

PEO_4_CF_3_COOLi + *x* wt.%	*σ*_0_ (S cm^−1^)	*E_A_* (eV)	*T*_0_ (K)
*x* = 0.0%	2.68 × 10^−2^	2.24 × 10^−2^	231.41
*x* = 0.3%	1.91 × 10^−1^	3.94 × 10^−2^	213.56
*x* = 0.6%	1.17 × 10^−2^	1.27 × 10^−2^	242.88
*x* = 2.0%	1.02 × 10^−1^	3.17 × 10^−2^	220.57
*x* = 4.0%	1.25 × 10^−2^	1.52 × 10^−2^	243.58
*x* = 8.0%	1.57 × 10^−1^	4.38 × 10^−2^	209.30
*x* = 12.0%	1.82 × 10^−3^	1.21 × 10^−2^	235.98

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
