# Peer review of "Composite Polymer Electrolytes Based on (PEO)4CF3COOLi and Multi-Walled Carbon Nanotube (MWCNT)"

_polymers, 2022, doi:10.3390/polym15010049_

Round 1
Reviewer 1 Report
The paper presents an interesting approach based on the Composites polymer electrolyte based in (PEO)4CF3COOLi and multi-walled carbon nanotube (MWCNT). However, the innovation of the current research work should be further highlighted and emphasized. At the same time, the authors should consider the following comments to greatly improve the quality of the paper.
1. In the abstract, add a final statement that highlights the importance of this research and its possible potentials. Also, introduce the problem in the initial lines of the abstract.
2. The introduction needs to be improved by relating to the mechanics of the studied materials and their mechanical characteristics. The references to be included are: 10.1016/j.polymertesting.2017.09.009, 10.1016/j.compstruct.2021.114698, 10.1177/0731684417727143, 10.1002/app.46770, 10.1016/j.porgcoat.2022.107015.
3. Kindly add a table that describes the main physical and chemical properties of the raw materials used in this study.
4. Were the preparation methods of the membranes described by the authors come in accordance with a certain standard or do they follow previous procedures?
5. What was the main reason for selecting the parameters: frequency range from 50 Hz to 5 MHz and a temperature range from 263 K to 338 K while conducting the impedance spectroscopy measurement of the samples?
6. Kindly justify the use of temperature range from 176 K to 430 81K in DSC testing, and justify the rate used 10 K/min.
7. Figure 7 has 9 different curves while five only appear in the legend. What are these 4 other curves representing? The axes of figure 7 need to be modified to match those of the other figures. This figure needs to be re-done.
8. Figure 8 has the same error as Figure 7. It needs to be re-done.
9. The conclusion needs to be modified to summarize the research outcomes in short statements with clear observations.
Reviewer 2 Report
In this paper, composites polymer electrolyte based in (PEO)4CF3COOLi and MWCNT were fabricated. And, their electrical, thermal, and structural properties were studied. The script was well written; however, the reviewer want to give some suggestions which may improve the quality of this paper.
(1) In the introduction, more literature reviews are required. There are many previous studies regarding polymeric composites using CNTs; however, they have some limitations to be used in real applications. Please find such limitations, and give some novel solutions in this paper.
(2) In addition, the dispersion of CNT in the polymeric composites is still an obstacle. So, please include more information about the dispersion problem. And, it is recommended to compare the used dispersion method compared to the previous studies. Please review the following paper, which used different dispersion methods. These can be used in this study to compare the dispersion methods.
-Facile Synthesis of Sprayed CNTs Layer-Embedded Stretchable Sensors with Controllable Sensitivity, Polymers (Basel). 13 (2021) 1–6.
-Electrical Stability and Piezoresistive Sensing Performance of High Strain-Range Ultra-Stretchable CNT-Embedded Sensors, Polymers (Basel). 14 (2022).
(3) For the composites using CNT, it is important to obtain the proper percolation threshold region. However, the investigations on the percolation threshold region have not been reported in the present script. So, it is highly expected to observe the percolation threshold and the electrical conductivity of the fabricated composites. The following paper can help you to understand the importance of the obtaining percolation threshold.
- Effect of carbonyl iron powder incorporation on the piezoresistive sensing characteristics of CNT-based polymeric sensor, Compos. Struct. 244 (2020) 112260.
Round 2
Reviewer 1 Report
The article can be accepted.
Reviewer 2 Report
The authors have revised the manuscript considering the reviewer's comments.
Thus, the reviewer think that it can be published in this journal.